# The Personal Resources of Successful Leaders: A Narrative Review

Kenneth Leithwood

Ontario Institute for Studies in Education, University of Toronto, Toronto, ON M5S 1V6, Canada; kenneth.leithwood@utoronto.ca

**Abstract:** Leaders' practices or overt behaviors are the proximal causes of leaders' effects on their organizations; they also dominate the research about successful leadership and often the content of leadership development programs, as well. But knowledge about those practices is, at best, a necessary but insufficient explanation for successful leadership and how it can be developed. This paper explores three categories of "personal leadership resources" that help explain why especially successful leaders behave as they do. These resources are often referred to as "dispositions", a term sometimes considered synonymous with traits, abilities, personal leadership resources and elements of a leader's personal "capital". The focus of this chapter is on three categories of resources (social, psychological and ethical) identified primarily through systematic research methods. For each category, the paper identifies the conceptual lens through which its dispositions are viewed and provides an explanation for how each of the specific dispositions within the category contributes to leaders' success. The paper also reviews a sample of evidence about contributions each disposition makes to leaders' success in achieving valued organizational outcomes. Implications for research and leader development are discussed in the concluding section of the paper.

**Keywords:** leadership antecedents; traits; dispositions; personal capital

## 1. Introduction

Leaders' practices or overt behaviors are the most direct causes of leaders' effects on their organizations; they also dominate research about successful leadership and often the content of leadership development programs, as well. But knowledge about those practices is, at best, a necessary but insufficient explanation for successful leadership and how it can be developed. This paper explores three categories of personal leadership resources that help explain why and how successful leaders behave as they do. "Personal leadership resources" (PLRs) is used here as a superordinate term encompassing such related terms as dispositions, traits, abilities and elements of leaders' personal "capital".

Two distinctly different methods have been used to justify attention to leaders' PLRs. One method, often associated with the selection of dispositions to be included in educational leadership standards and educational leadership development programs, for example [1], entails some form of logical analysis and/or practitioner judgement. PLRs justified using only this method often have high levels of face validity but unknown levels of predictive validity. The second method for justifying attention to a PLR includes the collection of systematic empirical evidence about the contribution of that PLR to some valued organizational outcome(s). Often based on relatively strong theory, PLRs justified using these systematic research methods typically demonstrate significant levels of predictive validity.

This narrative review paper examines three categories of PLRs identified primarily through systematic research methods; although leaders in schools and districts are the paper's target readers, this research has been conducted in a wide range of organizational contexts. The three categories of PLRs include key sets of interpersonal skills (social resources), closely related psychological dispositions or traits (psychological resources)

and strongly held personal beliefs leaders use as guides to morally defensible decision-making (ethical resources). For each of these categories of PLRs, the paper identifies the conceptual lens through which its specific resources are viewed, provides an explanation of how the resource contributes to leaders' success and cites examples of the evidence about contributions the resource makes to leaders' success in achieving valued organizational outcomes. An earlier, more limited account of these and other personal leadership resources can be found in Leithwood (2012) [2].

The three categories of personal resources examined in this paper are not exhaustive. Most obviously, they do not include what is typically referred to as "intellectual" or "cognitive" resources, such as, for example, academic intelligence, domain-specific problem-solving expertise and systems thinking. Cognitive resources such as these account for significant differences in leaders' emergence and performance [3] and will be the focus of a subsequent paper.

## 2. Social Resources

Among the terms used as synonyms for social resources, the three examined in this section come closest to illustrating the meaning of "capacities" or "abilities". The underlying theoretical and empirical foundation of these social resources can be found in the literature about social and emotional intelligence (SEI). Considerable evidence continues to link leader effectiveness to perceptions of leaders' SEI [4,5]. Much of this evidence builds on Salovey and Mayer's (1990) [6] original conception of emotional intelligence and Goleman's [7] popularization of the concept.

Although research about the relationship between school leaders' social resources and their success is still quite modest, evidence from other sectors points to a moderately strong relationship. Furthermore, the strength of this relationship will often depend on the extent of emotional labor associated with a job. Emotional management capacities, examined in this section of the paper, are likely to be strongly related to performance [8] in jobs requiring significant emotional labor. School leadership (and teaching) undoubtedly qualify as a '10' on the emotionally laborious scale.

Leaders' social resources contribute to valued organizational outcomes by promoting leader empathy, nurturing demonstrations of trust and confidence, keeping colleagues informed and showing appreciation for colleagues' ideas and recognition of their accomplishments.

Capacities captured in the SEI conceptual net encompass leaders' abilities to understand and manage the feelings, thoughts and behaviors of themselves and their colleagues in interpersonal situations and to act appropriately on that understanding. These social resources reflect the dimensions of emotional intelligence included in Salovey and Mayer's [6] original model (also see [9]).

### 2.1. Perceiving Emotions

Reflecting Salovey and Mayer's "social appraisal" dimension, perceiving emotions includes the ability to detect, from a wide array of clues, one's own emotions (self-awareness) and the emotions of others. Leaders with this ability are able to recognize their own emotional responses and how those emotional responses shape their focus of attention and influence their actions. They are also able to discern the emotions being experienced by others from, for example, tone of voice, facial expressions, body language and other verbal and non-verbal information.

Such empathetic capacities for the discernment of emotions being experienced by others, while extraordinarily valuable for leaders, need to be tempered by the chances of misinterpreting others' feelings, perhaps by extension of leaders' own sense of how they would feel in the others' situation. As Young [10] argues, empathetic dispositions need to be supplemented with engagement in respectful, reflective conversations to find out if what leaders' sense is accurate.

## 2.2. Managing Emotions

This social resource reflects Salovey and Mayer's [6] "emotional regulation" dimension. Leaders with this capacity manage their own and others' emotions, including the interaction of emotions on the part of different people in pairs and groups. Emotion management may promote positive affect and confidence in followers expressing and generating new ideas. Leaders with this ability are able to understand the reasons for their own intuitive, emotional responses and are able to reflect on the potential consequences of those responses. Leaders displaying this ability are also able to persuade others to be more reflective about their own intuitive, emotional responses and to consider the potential consequences of those responses.

## 2.3. Acting in Emotionally Appropriate Ways

Leaders with this social resource are able to respond to the emotions of others in ways that support the purposes for interacting with those others; they are able to exercise a high level of cognitive control over which emotions are allowed to guide their own actions and to assist others in acting on emotions that best serve the interests of those others. This social resource reflects the "emotional utilization" dimension of Salovey and Mayer's [6] theory.

A significant accumulation of meta-analytic reviews now provides a robust body of empirical evidence about the positive effects on valued organizational outcomes of the three social resources (perceiving, managing, acting) of leaders considered together. These reviews demonstrate, for example, the contribution of leaders' social resources to subordinate ratings of leaders' effectiveness [11], leaders' constructive conflict management practices [12], as well as leaders' use of demonstrably effective approaches to leadership [13–15]. Recent meta-analytic reviews also demonstrate the positive effects of leaders' social resources on employee task performance, organizational citizenship behavior [16] and job satisfaction [14,16].

It is worth noting that the social resources examined in this section of the paper share a common history in much earlier efforts to theorize leadership carried out at Ohio and Michigan State universities in the 1950s and 1960s. These studies situated "relationship building" among the two or three most important dimensions of effective leadership. According to these studies, effective leaders demonstrated "consideration" for their colleagues, for example, by acting in a friendly and supportive manner, showing concern for and looking out for their welfare.

Contemporary leadership theory continues to emphasize the importance of leaders' social resources. For example, transformational leadership theory assumes leaders' SEI abilities by including "individualized consideration" and "inspirational motivation" [17] among its categories of leadership practices. Leader–member exchange theory (LMX) [18] also assumes that leadership effectiveness depends on the ability to build differentiated relationships with each of one's colleagues, relationships that reflect their individual needs, desires and capacities.

*In sum*, the social resources needed to develop and sustain good working relationships have long been acknowledged as fundamental for leaders in almost all organizational contexts, and the importance of these resources increases with an organization's interpersonal intensity and the demands such intensity places on its leadership [19].

## 3. Psychological Resources

The common foundation of the six psychological resources examined in this section can be traced to a line of theory and research begun in the late 1990s [20,21], referred to as *positive psychology,* and its variants (positive organizational psychology (POP) and positive organizational behavior (POB)). Positive psychology is the science of "positive subjective experience, positive individual traits [and states], and positive institutions" [22]. While much of the mainstream organizational literature aims to unpack and respond to "problems", positive psychology aims to shine a light on "positively oriented human

resource strengths and psychological capacities that can be measured, developed, and effectively managed for performance improvement in today's workplace" [22] (p. 59).

The resources discussed in this section are sometimes referred to as psychological "states" rather than "traits" in order to distinguish differences in their malleability. Psychological traits are relatively stable, while states are much more malleable. Consistent with the priorities of positive psychology, states "can be measured, developed and effectively managed for performance improvement in the workplace and have significant practical implications for managers" [23] (p. 210).

A primary reason for highlighting psychological resources is the complexity of many leaders' jobs, perhaps especially school leaders. Complex jobs feature higher than average amounts of ambiguity, risk and uncertainty about achieving the outcomes for which many schools and their leaders are held accountable. As leaders' challenges become increasingly complex, there is an increasing drain on their psychological resources. Well-developed positive psychological resources allow leaders to cope productively in the face of high levels of complexity without giving up, experiencing excessive strain or becoming burnt out.

Four of the psychological resources included in this section of the paper make especially significant contributions to leader initiative, creativity and responsible risk-taking behavior [24–26]. Leaders are unlikely to deviate from well-established practices in order to improve their schools unless they believe they have a very good chance of being successful. Hope, optimism, self-efficacy and resilience make significant contributions to leaders' responsible risk-taking and eventual success [26]. When leaders possess high levels of all four of these psychological resources, they make an especially large contribution to leaders' initiative or "proactivity" [27,28]. Proactivity, in turn, is positively associated with leaders' work performance and the extent to which leaders have productive social relationships with their organizational colleagues. For example, Avey et al.'s [29] meta-analysis found significant effects of these four states on job satisfaction, organizational commitment and sense of well-being at work.

While psychological resources are clearly related, they are not emotions as, for example, the enjoyment, pride, frustration and anxiety included in Chen's [30] measure of leaders' emotions. Berkovich et al. define emotions as "affective experiences which . . . emerge when one perceives events or situations to have personal significance because they harm or promote oneself or one's goals" [31] (p. 130). Emotions can be fleeting, quickly changing in nature and highly dependent on the shifting contexts in which leaders find themselves. In contrast, psychological resources are more stable and will often be the basis on which emotions are determined. Consider, for example, differences in the emotional responses of two principals (A and B) about meeting with a parent who is very dissatisfied about how their child is being taught mathematics. Principal A has low levels of self-efficacy (a psychological resource) about her understanding of math instruction in the school, and so feels *anxious*, *nervous* and perhaps even *threatened* by the prospects of meeting with the parent. Principal B, with robust levels of self-efficacy or confidence about this matter, *enjoys* the prospect of meeting the parent because he views it as a prime opportunity to better explain to a parent the school's approach to math instruction. As this example demonstrates, leaders' psychological resources offer a more robust orientation than emotions for research aimed at explaining variation in leaders' success and a more promising focus for leadership development.

### 3.1. Proactivity

Proactivity is a motivational resource predisposing one toward initiating "future-oriented action to change and improve the situation" [32] (p. 636). Leaders who are proactive aim to bring about meaningful change in their organizations: they identify opportunities and act on them, show initiative and persevere in their efforts to bring about such change. Proactive leaders also transform their organization's mission, find and solve problems and take it upon themselves to have an impact on the world around them [33].

Evidence about the effects of leaders' proactivity demonstrates its contribution to the likelihood of being perceived as a leader, as well as to the achievement of organizational goals [33]). The overt actions taken by proactive leaders are stimulated or supported by other psychological resources, including optimism, self-efficacy and resilience.

### 3.2. Optimism

This psychological resource has been defined as "both a positive future expectation open to development . . . and an explanatory/attribution style interpreting negative events as external, temporary, and situation specific, and positive events as having opposite causes (i.e., personal, permanent, and pervasive)" [29] (p. 130). Optimistic leaders habitually expect to succeed in their efforts to address challenges and confront change in both the present and the future. Optimistic leaders habitually expect good things to result from their initiatives, while pessimistic leaders habitually assume that their efforts often will be ineffective. When the expectations of optimistic leaders are not met, they pursue alternative paths to accomplish their goals. Optimistic leaders have productive responses to stress, do not give up easily and act to overcome sources of stress.

Leaders' optimistic expectations do not necessarily extend to their organizations as a whole. Rather, optimistic leaders are realistic as well as optimistic. They expect their efforts to be successful in relation to those things over which they have direct influence or control but not necessarily to be powerful enough to overcome negative forces in their organizations over which they have little or no influence or control. Kluember, Little and DeGroot conclude from their study that ". . . optimism . . . is a potentially powerful indicator of important organizational outcomes, even after controlling for the effects of positive and negative affect" [23] (p. 209). Optimism is also positively associated with job satisfaction, work happiness, organizational commitment and performance [34].

### 3.3. Self-Efficacy

Both optimism and self-efficacy contribute to the likelihood of a leader continuing to strive for success even in the face of initial failure. Unlike optimism, however, efficacy's contribution is both ability- and performance-based [35]. Self-efficacy is a belief about one's own ability to perform a task or achieve a goal. Efficacious leaders believe they have the ability to solve whatever challenges, hurdles or problems that might come their way in their efforts to help their organizations succeed.

Self-efficacy beliefs contribute to leaders' success through their influence on leaders' choices of activities and settings and can affect coping efforts once those activities have begun [36]. Efficacy beliefs determine how much risk people will take, how much effort they will expend and how long they will persist in the face of failure or difficulty; the stronger the self-efficacy, the longer the persistence. Leader self-efficacy is likely the key cognitive variable regulating leader functioning in a dynamic environment and has a very strong relationship with leaders' performance [37–39]. Sun, Chen and Zhang's (2017) meta-analysis [5] reported an outsized relationship between self-efficacy and transformational approaches to leadership.

### 3.4. Resilience

Resilience, the ability to recover from or adjust easily to misfortune or change, is significantly assisted by high levels of self-efficacy but goes beyond the belief in one's capacity to achieve in the long run. At the core of resilience is the ability to "bounce back" from failure and even move beyond one's initial goals while doing so [40]. Resilient leaders thrive in the challenging circumstances commonly encountered by school leaders. They recognize the destructive consequences of some setbacks but often use those setbacks as motivation to move beyond their initial circumstances. As Paglis explains, while "hope and optimism best apply to situations that can be approached with a plan . . . resilience recognizes the need for flexibility, adaptation, and even improvisation in situations predominantly characterized by change and uncertainty. . .". The resilient leader "searches for and

finds meaning despite circumstances that do not lend themselves to planning, preparation, rationalization, or logical interpretation" [35] (p. 780).

Along with hope, optimism and self-efficacy, leader resilience is "positively related to a variety of employee attitudinal, behavioral, and performance outcomes" [28] (p. 37), including, for example, organizational culture and organizational commitment.

### 3.5. Hope

Some treatments of hope distinguish between hope as a response to some immediate event or challenge (dispositional hope) and a more predictable response to many events or challenges over time (state hope). State hope is "a positive motivational state that is based on an interactively derived sense of successful (1) agency (goal-directed energy) and (2) pathways (planning to meet goals)" [29] (p. 130).

Leaders with hope perceive themselves as able to initiate and maintain actions required to achieve some desirable goal. Those leaders also are able to imagine ways in which to achieve that goal. These two components of hope are "iterative, additive, and positively related, but are still conceptually distinct constructs" [41] (p. 26). Leaders possessing this psychological resource persistently work to control their circumstances and engage in planful behaviors aimed at addressing the many challenges they encounter in those circumstances.

Peterson and Luthans's study found that "high-hope" leaders had significantly better work unit financial performance, subordinate retention and satisfaction outcomes than "low-hope leaders" [41] (p. 29). Reichard et al.'s [42] (2013) extensive quantitative meta-analysis also found significant effects of employee hope on their work performance and sense of well-being. Transformational leadership practices are associated with improvements in employee hope.

### 3.6. Humility

While the quantity of empirical evidence about the organizational effects of leader humility is still modest, humility is an important feature of some well-developed leadership models, including authentic leadership, distributed leadership and some versions of transformational leadership. Leaders perceived to be genuinely humble build an organizational climate of psychological safety and democracy, which encourages the contribution of all members to their collective learning and goal achievement. Genuine humility is also likely to help build trust among group members, a key condition for authentic communication. Other important psychological dispositions among leaders' colleagues (e.g., hope, optimism) are positively influenced directly by leader humility and indirectly through leader effects on colleagues' humility [43]. Evidence from studies by Owen and colleagues indicates that "humble leaders foster learning-oriented teams and engaged employees, as well as promote job satisfaction and employee retention" [44] (p. 1533).

While leader humility is not uniformly beneficial [45], leaders who are perceived to be genuinely humble, rather than simply giving the impression of being humble, also encourage humility among their organizational colleagues. In the context of learning-oriented teams, for example, such humility manifests itself as

"*A willingness to evaluate themselves without exaggerating their accomplishments either positively or negatively; an appreciation for the unique strengths and contributions of those with whom they work; and openness to new ideas, feedback, and advice*" [46] (p. 641)

*In sum,* the six psychological resources examined in this section serve at least three critical purposes. Primarily served by a proactive disposition, one purpose is to nurture a strong urge to make things better, to not settle for the status quo. Indeed, proactivity is at the heart of what leadership means in many organizations. A second purpose for these psychological resources is to provide leaders with the stamina needed to cope effectively in interpersonally demanding environments for extended periods of time in order to make things better; *hope, optimism, self-efficacy* and *resilience* serve this purpose especially well.

Finally, one psychological resource, *humility*, encourages other organizational members to adopt a positive view of their own abilities to contribute productively to their organization. Leader humility encourages others in the organization to exercise their own capacities, to continue their own professional learning and to do this in in collaboration with their colleagues.

## 4. Ethical Resources

A very high proportion of decisions faced by leaders are values-laden. This category of resources includes the moral values that motivate leaders when decisions are about the right thing to do. Leaders with well-developed ethical resources behave in accordance with their moral values as often as possible in their own circumstances. "As often as possible" means doing the right thing for the leaders' organization, its employees and those people the organization aims to serve while acknowledging that doing the right thing for all three of these entities at the same time can sometimes be a herculean challenge. Different and sometimes conflicting values regularly come into play. Popular admonishments for leaders to be "authentic" do not help. Much more than just the leaders' own values need to be considered in their decision-making.

This section is limited to identifying leaders' moral values linked to desirable organizational outcomes, but it does not attempt to address the complex task leaders face when deciding among competing values, all of which may be good for some but at least not optimal for others. *Ethical Leadership Theory* is used as the theoretical foundation on which to construct an account of leaders' ethical orientations. According to such theory, ethical leaders do what is morally right and help their colleagues do the same [47]; they hold values that are both other-centered and reflect high standards of morality. When leaders' own behaviors are perceived to reflect those values and when leaders create infrastructures that reward the expression of those values [48], organizational members are persuaded to act according to those values, as well.

The core values of an organization, manifest in ethical leaders' practices, are central features of an organization's culture. This organization-wide sharing of values increases members' efforts to ensure the success of their organization and enhances the attractiveness of the organization to those it aims to serve. Brown and Trevino's comprehensive review of research suggests, more explicitly, that ethical leaders are the following:

> "*Honest, caring, and principled individuals who make fair and balanced decisions. Ethical leaders also frequently communicate with their followers about ethics, set clear ethical standards and use rewards and punishments to see that those standards are followed. Finally, ethical leaders do not just talk a good game—they practice what they preach and are proactive role models for ethical conduct. Ethical leaders are characterized as honest, caring, and principled individuals who make fair and balanced decisions. Ethical leaders also frequently communicate with their followers about ethics, set clear ethical standards and use rewards and punishments to see that those standards are followed*" [49] (p. 597)

The values described in this quotation are examples of values widely believed to be central features of individual and organizational morality in most Western contexts. These widely endorsed "fundamental" moral values are necessary components of ethical leadership in virtually all organizational contexts. Most such contexts, however, demand that leaders consider additional "professional" values. Values related to equity and social justice are, without doubt, the most notable of those professional values that contemporary educational leaders must consider in their decision-making; while they are the only professional values discussed here, many more will often come into play depending on leaders' contexts.

### 4.1. Fundamental Moral Values

Values alluded to in the quotation above by Brown and Trevino [49] include honesty, trustworthiness, fairness and care about the welfare of others. While all four of these values count as "fundamental", a growing body of theory and evidence, especially in

education, highlights the primary importance of *care* [50] along with three closely related values—benevolence, virtue and altruism.

Leaders in schools, for example, who value *care* are attentive to the needs of staff and students, act on behalf of staff and students in a selfless way and are viewed as open, transparent and genuine in their beliefs [51,52]. This conception of care encompasses most of what is typically included in research on benevolence, virtue and altruism. Cameron, Bright and Caza describe leaders who are *benevolent* as working to develop moral goodness and social betterment within their organizations without the expectation of personal gain. The same authors describe *virtuous* leaders as those who aspire to be "at their very best" [53] (p. 767) both individually and professionally; these aspirations are manifest in both individual and collective actions, organizational culture and in other processes that develop and promote moral behavior across the organization. Finally, *altruistic* leaders are described by Sosik et al. [54] as selflessly providing help to individual members of their organizations whenever needed.

Evidence across many organizational sectors associates care and its close relatives (benevolence, virtuousness and altruism) with such outcomes as greater innovation, improved customer retention, decreased employee turnover and greater profitability [53]. Other outcomes include improved job performance [54] and higher levels of organizational commitment and employee job satisfaction [55]. Such satisfaction and commitment, as well as employee trust in leaders, develop when leaders help to ensure that employees find meaning in their work, foster their capacities to do their jobs well and allow them significant opportunities for determining how to carry out that work [47]. Louis and Murphy [52] (2017) concluded, from their large-scale empirical study in schools, that teacher perceptions of their principals as strongly valuing care are significantly related to student achievement and to increased efforts by teachers to better serve the needs of marginalized students.

### 4.2. Professional Values: Equity and Social Justice

The term "equitable" acknowledges that some students need more of the school's resources and opportunities than others to achieve the same levels of success. "Equitable" is often used as a superordinate term encompassing such related concepts as social justice and inclusion. Most of the authors contributing to the 27 chapters of the *Handbook for Ethical Educational Leadership* [56] portray ethical leadership as heavily focused on improving equity in schools.

This set of values has been awarded special attention as society increasingly acknowledges the economic, structural, social and other challenges faced by many non-majority members of the population through no fault of their own. In education, this acknowledgement is reflected in widespread policies designed to "even the playing field" for all students and similarly widespread efforts to better prepare educators with the sensitivities and skills needed to achieve the goals of those policies. Sources of inequity demanding educational leaders' attention by these policies include, for example, various forms of disability, language, culture, sexual orientation, ethnicity, race, social class, color and poverty; individual students frequently experience multiple sources of inequity at the same time.

Results of a recent review of equitable leadership practices and dispositions [57] include a large handful of more specific values associated with school leaders aiming to advance equity and social justice, including the following: commitment to achieving greater equity for students; beliefs in the integrity of Indigenous ways of knowing and knowledge [58]; and resistance to dominant narratives based on colonizing assumptions and a robust sense of self-efficacy about improving social justice in one's school including a personal sense of "being right", but being humble about one's accomplishments [59]. Leaders strongly committed to social justice also have the courage required to disrupt and resist sources of bias and racism among staff and other stakeholders, as well as engage in difficult conversations with staff and other stakeholders about bias, deficit perspectives and more. Such courage is also exercised in the face of community and accountability pressures [60].

*In sum*, leaders with significant ethical resources are committed to finding the morally most defensible courses of action in the face of competing values and other interests; they also have the capacity to justify those courses of action to others.

## 5. Conclusions

There is a considerable body of research now describing the practices of successful leaders and how those practices influence important organizational outcomes. Less attention has been devoted to the antecedents of those practices (but see [5]), including the personal resources of successful leaders. These resources, however, play a significant role in explaining why some leaders chose to enact the successful practices documented by so much research and how they enact those practices in their own contexts. These resources also help explain variations in the emotional responses of leaders to very similar conditions.

Three categories of personal leadership resources have been explored in this paper—social, psychological and ethical resources (leaders' cognitive resources will be addressed in a subsequent paper). For each category, the paper has identified a foundational theory, explained how the multiple resources in each category contribute to leader success and reviewed a representative sample of evidence about the immediate and longer-term organizational outcomes associated with each of those resources. The paper concludes with four recommendations.

The first recommendation concerns future research. While there is a substantial body of well-designed empirical research about many of the personal leadership resources touched on in the paper, the amount of that research carried out in the educational sector is quite small. Much more is needed to confirm or modify the results of research conducted in other sectors. A large-scale study now underway by faculty at University College London (UK) provides one example of the form such research might take. This study includes, among other things, the influence of headteachers' psychological resources on the retention decisions of their teachers [61].

A second recommendation concerns leadership development. While the paper has not touched on how leadership resources can be developed, the evidence does indicate that for most of these resources, this is clearly possible—and in contexts common to many leadership development initiatives (e.g., see Rafferty and Griffin [62] on developing self-efficacy and Parker et al. [32] on developing proactivity). Leithwood [63] provides a comprehensive review of pedagogical strategies for developing leaders' personal resources, as well as other skills and knowledge. Focused on the development of social resources, for example, evidence suggests that strategies for increasing leaders' sensitivity to their own and others' emotions include coaching and feedback for recognizing the emotions of self and others from facial and voice cues, as well as awareness of one's own body states relating to emotions. Also recommended is coaching about how to adjust the overt expression of leaders' own emotions so they are appropriate to the circumstances and how to draw on positive moods to enhance creative thought [64].

The third and closely related recommendation concerns the diagnosis of individual leaders' personal resources as part of leadership development experiences, for example. A substantial amount of work has been completed on how such diagnosis might be carried out; see, for example, Snyder et al. [65] on assessing hope, Scheier and Carver [66] on assessing optimism and Connor and Davidson [67] on assessing resilience. So, a good start has been made toward the creation of tools needed to diagnose the status of individual leader's personal resources. Systematic formative assessment of personal leadership resources should be a more significant starting point for both pre-service and in-service leadership development programs than is presently the case.

A final recommendation is aimed at the senior leaders with whom school leaders work. While personal leadership resources can be developed, they can also be eroded. Frederickson [68] (p. 330) refers to many of the PLRs discussed in this paper as "*enduring* personal resources". But the term "enduring" needs to be understood as mostly *useful when available*, not as *able to withstand most of the challenges* and issues faced by school leaders.

School leaders assigned to new schools can encounter novel circumstances likely to erode, at least temporarily, their sense of efficacy as competent leaders. Leaders' optimism and hope are likely to be eroded over time without occasional successes, for example, in bringing staff together in support of an important school goal. So, this final recommendation is for senior district leaders. Cherish and nurture the personal resources your school leaders bring to their jobs. At the very least, do not challenge or erode them by assigning your school leaders responsibilities for accomplishing impossible goals (e.g., excessive numbers of new mandates to be implemented that are largely incoherent). At best, allow your school leaders considerable discretion about school-level decisions about how district mandates are to be implemented and what priorities make the most sense in their own schools. District leaders might also consider the nature of the challenges likely to be encountered by a principal reassigned to a new school and, with that principal, identify additional training opportunities to help prepare the principal in advance of taking on the new school.

**Funding:** This research received no external funding.

**Data Availability Statement:** Not applicable.

**Conflicts of Interest:** The author declares no conflict of interest.

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
