# Peer review of "The Personal Resources of Successful Leaders: A Narrative Review"

_education, doi:10.3390/educsci13090932_

Round 1

Reviewer 1 Report

This piece does a very nice job of organizing existing literature into some categories that shows how it is related and relevant. I am not expert in all of the research overviewed here, so I cannot comment on the the cogency of a specific citation or if an important reference is missing. As a professor of ed leadership, I can remark that I find this pulling together of material is very helpful for considering my institution's preparation program, the discussion in the field (i.e., lit and standards, etc) of dispositions. Because of the lack of clarity around their measurement and development, and the need for further research, it seems the dispositions conversation has dried up. This article could do more to help prompt the use of this synthesis it presents. I urge the author(s) to do some further elaboration in the conclusion section, and also to add some implications.

Please discuss recommendation 1 to 3 further as to how the field might do more with these suggestions. The author notes in recommendation 3 that a substantial  amount of diagnostic tool development has been done, yet only a couple of examples are provided. It would be a great service and significantly extent the usefulness of this piece to provide further information on such tools. This could become a new sub-section within each of the three main sections above, or more information provided with the recommendation. The implication for leadership development programs is mentioned, but also could provide further elaboration to spur readers from the professoriate to act. 

The last recommendation, too, might elaborate a bit on what cherish and nurture might look like. 

Because I was interested in looking at several of the referenced pieces, I noticed some things about the references.  The author cited himself several times in the article with his name and then inserted those references as author, year, but in the alphabetically appropriate spot, rather than at the beginning of the list, which isn't in the spirit of de-identification for peer review.  Also, the referenced Sun, Chen and Zhangh (2017) piece is missing from references list. I did not systematically cross-check to see if other refs are missing, but the author should. 

Reviewer 2 Report

The paper reads beautifully and is well-structured. It is really very informative and comprehensive in nature and adds to the existing literature, albeit not to the fullest extent. I would expect a bit more towards the end, as regards to answering the "so what" question in more practical and applicable terms. Still, it is a paper very much worth reading and publishing.
